# Unleashing the Creative Mind: Language Model As Hierarchical Policy For Improved Exploration on Challenging Problem Solving

## Abstract

Large Language Models (LLMs) have achieved tremendous progress, yet they still often struggle with challenging reasoning problems. Current approaches address this challenge by sampling or searching detailed and low-level reasoning chains. However, these methods are still limited in their exploration capabilities, making it challenging for correct solutions to stand out in the huge solution space. In this work, we unleash LLMs' creative potential for exploring multiple diverse problem solving strategies by framing an LLM as a *hierarchical policy* via in-context learning. This policy comprises of a *visionary leader* that proposes multiple diverse high-level problem-solving tactics as hints, accompanied by a *follower* that executes detailed problem-solving processes following each of the high-level instruction. The follower uses each of the leader's directives as a guide and samples multiple reasoning chains to tackle the problem, generating a solution group for each leader proposal. Additionally, we propose an effective and efficient *tournament-based approach* to select among these explored solution groups to reach the final answer. Our approach produces meaningful and inspiring hints, enhances problem-solving strategy exploration, and improves the final answer accuracy on challenging problems in MATH.

## 1 Introduction

Large language models (LLMs) (Brown et al., 2020; Chowdhery et al., 2022; Touvron et al., 2023; OpenAI, 2023) have demonstrated remarkable potential across a myriad of disciplines such as common sense understanding (Hendrycks et al., 2020; Srivastava et al., 2022) and code generation (Chen et al., 2021; Li et al., 2023). Yet, LLMs often struggle with challenging reasoning tasks, such as writing mathematical proof and solving advanced mathematics problems. These tasks are inherently creative, as the path to a solution isn't immediately evident, requiring the exploration of numerous problem-solving tactics before discovering a successful path towards the end goal.

While recent works have investigated enhancing LLMs' exploration ability in problem solving through sampling and search (Wang et al., 2022; Yao et al., 2023; Besta et al., 2023), these approaches still exhibit considerable limitations. Before we describe such limitations, let's think of *how humans approach mathematical proofs*: one typical methodology is that we begin by connecting the target proof statement to our prior experiences such as proofs with similar routines (e.g., divide-and-conquer) or relevant techniques (e.g., root-finding). From this reservoir of knowledge and familiarity, **humans try multiple "high-level" proof tactics and directions that possess the potential to reach the goal, and subsequently develop detailed, "low-level" proof details based on these directions**. *It should be noted that the quality of "high-level" strategies and thinking processes can exert a substantial impact on the effectiveness, efficiency, and likelihood of successfully solving these problems, as illustrated in Tab. 1.* In cognitive science, such advanced higher-order thinking skills are referred to as *Metacognition* (Davidson et al., 1994; Metcalfe & Shimamura, 1994; Berardi-Coletta et al., 1995). It is widely acknowledged that metacognition ability leads people to effective problem-solving strategies and successful task completion (Swanson, 1992; Alexander et al., 1995).

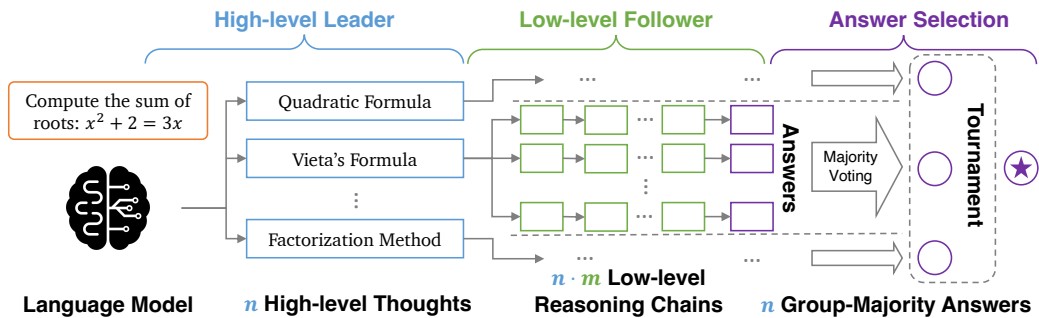

Figure 1: Overview of our approach, which frames language models as a **hierarchical policy** for exploration. The **visionary high-level leader policy** connects the target problem with the language model's knowledge, proposing multiple diverse tactics and hints for exploration. The **low-level follower policy** leverages these hints as in-context guidance to execute detailed problem-solving processes. Finally, we employ an **effective and efficient tournament-based approach** to select the desired reasoning chains and reach the final answer.

*Compared to human exploration of complex problem solution spaces, the aforementioned sampling and search methods in NLP have primarily focused on delving into the detailed, "low-level" reasoning steps, often overlooking the "high-level" strategies and cognitive processes.* We, therefore, aspire to unleash LLMs' potential for creative exploration of high-level tactics and hints, enabling them to tackle challenging reasoning problems with similar ingenuity and proficiency as humans.

To this end, we draw inspiration from the concept of a "hierarchical policy" in the decision-making literature (Bacon et al., 2017; Li et al., 2017; Kulkarni et al., 2016), and we propose to define LLM as a **hierarchical policy** for problem solving, which consists of a visionary high-level leader policy and a low-level follower policy. In our framework, the high-level leader establishes connections between the target problem and the LLM's extensive knowledge and prior problem-solving experiences. It leverages this information to propose various high-level problem-solving tactics and directions for exploration. The low-level follower policy then utilizes each of these hints as an in-context guidance throughout the detailed step-by-step problem-solving processes. Furthermore, we desire implementations of this idea to be achieved through minimal effort. Indeed, this can be achieved by leveraging off-the-shelf pretrained language models and in-context learning. Finally, after we obtain an array of diverse reasoning chains through LLM's creative exploration process, we propose an effective and efficient tournament-based method to select among these chains to arrive at the final answer. We illustrate an overview of our approach in Fig. 1.

Experimentally, we demonstrate that our high-level leader policy is capable of exploring and producing meaningful and inspiring hints and guidance for the low-level follower policy, thereby making it easier for correct reasoning chains and answers to stand out. Our reasoning chain selection approach effectively identifies desired reasoning chains, enhancing the final answer accuracy on challenging mathematical reasoning tasks. Our key contributions are as follows:

1. To effectively explore expansive solution spaces in complex problems, we propose framing large language models (LLMs) as a **hierarchical policy, encompassing both "high-level" and "low-level" cognitive processes**, facilitating a more efficient exploration of distinct high-level ideas.

2. Within our hierarchical policy, we present two effective approaches for the visionary high-level leader policy to generate a diverse set of tactics and hints that guide the low-level follower policy during exploration.

3. We propose an **effective and efficient tournament-based approach** for selecting desired reasoning chains among those generated during exploration, facilitating the attainment of the final answer.

4. Experimentally, we demonstrate that our approach produces high-level hints and guidance that are meaningful and inspiring, enhances problem-solving strategy exploration, leads to better discovery and visibility of correct solutions, and improves the final answer accuracy on challenging problems in the MATH dataset.

**Question**: For how many integer values of $a$ does the equation $x^2 + ax + 5a = 0$ have integer solutions for $x$?

**Ground Truth Solution (High level approach: Vieta's Formulas + Factorizing Trick)**:
Suppose the roots of the quadratic are given by $m$ and $n$. Note that $(x - m)(x - n) = x^2 - (m + n)x + mn$ $= x^2 + ax + 5a$, and setting coefficients equal, it follows that $m + n = -a$ and $mn = 5a$ (This also follows directly from Vieta's formulas.) Notice that the $a$ can be canceled by either dividing or noting that $0 = 5a$ $+5 \cdot (-a) = mn + 5(m + n)$. Simon's Favorite Factoring Trick can now be applied: $mn + 5m + 5n + 25$ $= (m + 5)(n + 5) = 25$. It follows that $m + 5$ and $n + 5$ are divisors of 25, whose pairs of divisors are given by $\pm\{(1, 25), (5, 5), (25, 1)\}$. Solving, we see that $(m, n)$ is in the set $\{(-4, 20), (0, 0), (20, -4), (-6, -30),$ $(-10, -10), (-30, -6)\}$. However, the two pairs of symmetric solutions yield redundant values for $a$, so it follows that the answer is $\boxed{4}$.

**Example Reasoning from GPT-4 (High level approach: Discriminant is a perfect square)**:
First, we note that the discriminant of a quadratic equation must be a perfect square for the roots of the equation to be integers. The discriminant of our equation $x^2 + ax + 5a = 0$ is $a^2 - 20a$. Setting this equal to $k^2$ for some integer $k$, we get $a^2 - 20a - k^2 = 0$. This is a quadratic equation in $a$, and its roots are $a = 10 \pm \sqrt{100 + k^2}$. Again, for $a$ to be an integer, $\sqrt{100 + k^2}$ must be an integer. Let $\sqrt{100 + k^2} = m$. Then $m^2 = 100 + k^2$ $\implies k^2 = m^2 - 100$. This means that $m^2 - 100$ must be a perfect square. The possible values of $m$ which satisfy this condition are $m = 10, 11, 12, 13, 14, 15, 20$. For each of these 7 values of $m$, we get 2 values of $a$ (one with the $+$ sign and one with the $-$ sign). However, we have to be careful with $m = 10$, because in this case $k = 0$, and we get only one value of $a = 10$. So, the total number of integer values of $a$ for which the equation $x^2 + ax + 5a = 0$ has integer solutions for $x$ is $2 \times 6 + 1 = \boxed{13}$.

Table 1: An example problem from the MATH dataset, which can be solved through two distinct high-level approaches: (1) Identify when the discriminant is a perfect square; (2) Apply Vieta's formulas and the factorization trick. *The first high-level approach does not effectively reduce the problem into a simpler form*, as solving for all $a$ where $a^2 - 20a$ is a perfect square remains a highly challenging task, which causes the problem solving process to become stuck. Conversely, *adopting the second strategy leads to an elegant and concise solution.*

## 2 RELATED WORK

**Reasoning and Exploration with Language Models.** Recent large language models (LLMs) have demonstrated remarkable potentials in solving complex reasoning tasks. A key strategy is to encourage LLMs to generate detailed step-by-step reasoning processes through in-context learning techniques, including but not limited to Chain-of-Thought (CoT) prompting (Wei et al., 2022) and numerous other approaches (Kojima et al., 2022; Zhou et al., 2022a; Zhang et al., 2022; Si et al., 2022; Wang et al., 2022; Zhou et al., 2022b; Liu et al., 2023; Zhou et al., 2023; Yao et al., 2022). For many challenging reasoning tasks such as mathematical problem solving, proof writing, and inductive reasoning, it is often challenging for LLMs to obtain the correct solution in a single attempt. Therefore, to further enhance LLMs' problem solving capabilities, it is highly beneficial to encourage LLMs to search and explore over diverse problem-solving strategies and reasoning steps. Recent works like Bostrom et al. (2022); Tafjord et al. (2021); Yang et al. (2022); Creswell et al. (2022) perform step-by-step search to construct deductive natural language proofs given premises. Wang et al. (2022) samples and explores multiple reasoning chains, then performs majority voting to obtain the final answer. Weng et al. (2022); Lightman et al. (2023); Ling et al. (2023) introduce verification filtering to the reasoning chain exploration process. Yao et al. (2023) and Besta et al. (2023) perform tree-based and graph-based search over reasoning steps with backtracking and refinement. However, much prior work limits reasoning exploration to specific and detailed reasoning steps, and the high-level strategies and thinking processes are often overlooked. In this study, we frame LLMs as hierarchical policies, enhancing problem-solving by exploring diverse high-level strategies and tactics.

**Hierarchical Policy.** The concept of hierarchical policy was originally proposed in reinforcement learning (RL) and imitation learning (IL) as a multi-level decision-making approach to tackle complex, long-horizon tasks (Bacon et al., 2017; Li et al., 2017; Kulkarni et al., 2016; Nachum et al., 2018; Gupta et al., 2020; Pertsch et al., 2021). In hierarchical RL, a high-level policy proposes latent features and subgoals that guide low-level policy actions. Prior work has also investigated enhancing the exploration abilities of hierarchical policies (Li et al., 2020; Gehring et al., 2021; Peng et al., 2022; Huang et al., 2023). However, few prior works has framed large language models as hierarchical policies through in-context learning to improve their exploration capabilities in problem solving, which is the main focus of our work.

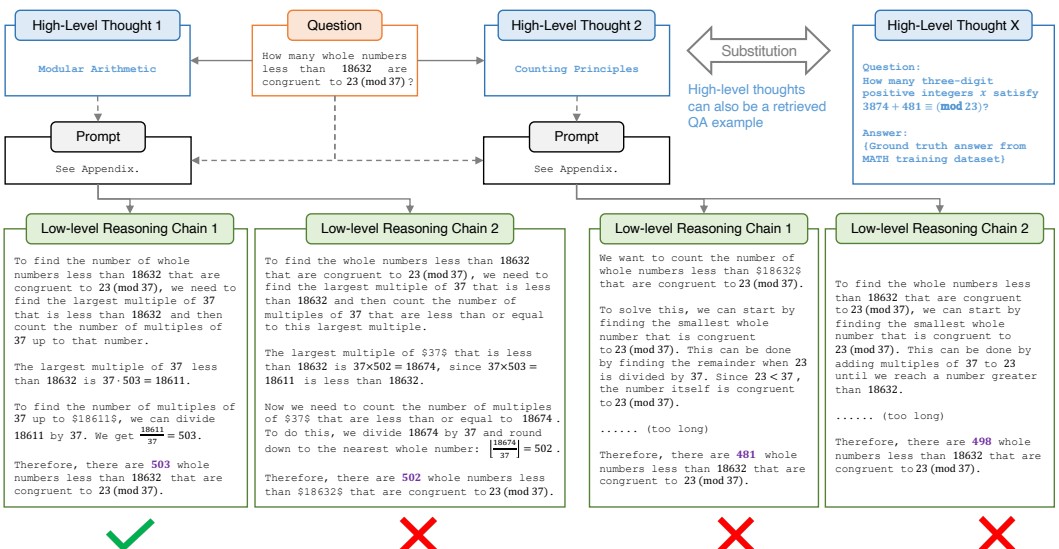

Figure 2: A detailed illustration of our approach that frames language models as a hierarchical policy for exploration (zoom in for better view). The high-level leader proposes diverse hints that guide the low-level follower in problem-solving exploration. These hints can take the form of concise techniques and theorems, or a set of retrieved similar problems and solutions. In this example, "Modular Arithmatic" is highly relevant to the target question, and the low-level follower successfully finds the correct answer in some generated reasoning chains. On the other hand, "Counting Principles" is irrelevant to the target question, and the low-level follower struggles to reach the correct solution. Later, we propose an effective and efficient approach to select the desired reasoning chains from those generated by the low-level follower.

## 3 LANGUAGE MODEL AS A HIERARCHICAL POLICY FOR EXPLORATION

A natural language reasoning process can be defined as a tuple $(Q, T, A)$, where $Q$ is the target question; $A$ is the ground-truth answer in the format of a number in math word problems or a statement to proof or conclude; $T$ is the set of locally-valid reasoning chains that reach the ground-truth answer, i.e., $T = \{\tau = (\tau_1, \tau_2, \ldots, \tau_s) : \text{valid}(\tau) = 1, \tau_1 = Q, \tau_s = A\}$. A large language model (LLM), denoted as $\pi$, takes a question $Q$ and a prompt $P$ as input to generate a step-by-step reasoning chain $\tau = \pi(P, Q)$ that attempts to solve the problem. In the quest to improve LLMs' exploration abilities in problem solving, much prior work focuses on exploring, sampling, and searching for specific reasoning steps (Wang et al., 2022; Yao et al., 2023; Besta et al., 2023). *Yet, these methods tend to neglect the higher-order cognitive processes inherent in human problem solving. Successful problem-solving often relies on a guiding strategy and hint, and overlooking this aspect could potentially lead to inefficient and ineffective exploration.*

To address these limitations, we propose to formulate LLM as a **hierarchical policy** $\pi = (\pi_{high}, \pi_{low})$ for problem solving through in-context learning. Following the convention of Markov Decision Process, $\pi$, $\pi_{high}$, and $\pi_{low}$ are probabilities over token sequences. The visionary high-level leader policy $\pi_{high}$ takes in a question $Q$ and a prompt $P_{high}$ as input and returns a set of possible high-level tactics and hints $H = \{h_1 \ldots h_n\}$, where $H \sim \pi_{high}(P_{high}, Q)$ (we emphasize that a sample from $\pi_{high}$ returns a *tactic set* instead of a single tactic; we will also use tactic / hint interchangably from here on). Then, the low-level follower policy $\pi_{low}$ utilizes *each* $h_i$ as an in-context guidance to execute specific problem-solving processes by sampling or searching reasoning chains, yielding a **group** of reasoning chains $T_i = \{t_{i,1}, \ldots, t_{i,m}\}$, where $t_{i,j} \sim \pi_{low}(h_i, Q)$.

To successfully instantiate our hierarchical approach, there are **two crucial design components** we need to address: **(1)** How to encourage the leader $\pi_{high}$ to generate appropriate tactics and hints $H$ that serve as effective guidance for the follower $\pi_{low}$; **(2)** Given groups of reasoning chains $\{T_i\}_{i=1}^n$ generated by $\pi_{low}$, how to effectively and efficiently choose the desired reasoning chains to obtain the final answer.

**Generating high-level tactics and hints for exploration.** Given a question $Q$, our goal is to encourage the leader $\pi_{high}$ to effectively establish the connection between $Q$ and relevant language model knowledge, from which it proposes high-level hints and directions that holds significant potential for reaching the goal. We would also like to limit irrelevant hints that potentially mislead the low-level policy, since previous work (Shi et al., 2023) has shown that language models can be brittle to irrelevant context. *We therefore propose **two approaches** for $\pi_{high}$ to generate high-level problem-solving tactics and hints* (an illustration in Fig. 2):

(a) **Prompt** an LLM, such as GPT-4, to generate a list of relevant techniques and concepts for a target question. We aim for these hints to be clear, concise, and easily interpretable, e.g., *"Angle Bisector Theorem"*, *"Minimization with Derivatives"*, such that they can serve as effective guidance for the low-level policy $\pi_{low}$. See Appendix for detailed prompts.

(b) Use a sequence-embedding language model, such as SentenceBERT (Reimers & Gurevych, 2019), to **retrieve** a set of relevant problems with their step-by-step solutions. Each relevant problem and solution then inspires $\pi_{low}$ to utilize similar tactics and strategies when exploring and generating reasoning chains.

*Probablistic Interpretation:* Next, we build a connection between our method and hierarchcial policies in the Markov Decision Process (MDP) framework, and we use the MDP framework to explain the improved exploration ability. To make it intuitive, we use Fig. 3 to illustrate the idea. In the low-level reasoning chain space $T = \{t_1, t_2, \ldots\}$ given $Q$, we group reasoning chains based on the high-level tactics they employ. For example, $t_{1,1}$ and $t_{1,2}$ both employ tactic $h_1$. Here, the size of the region corresponds to the marginal conditional probability of $h$ given $Q$ when we sample the low-level reasoning chain **without** providing a high-level tactic prompt. We denote this marginal probability as $\Pr(h|Q)$. Note that the tactic with the highest $\Pr(h|Q)$ may *not* necessarily lead to the correct reasoning chains. This should not be counter-intuitive, especially for hard math problems that require out-of-the-box thinking. In practice, for a specific math question that receives an incorrect answer from GPT-3.5/4, we have observed that the generated reasoning chains frequently rely predominantly on a *single* tac-

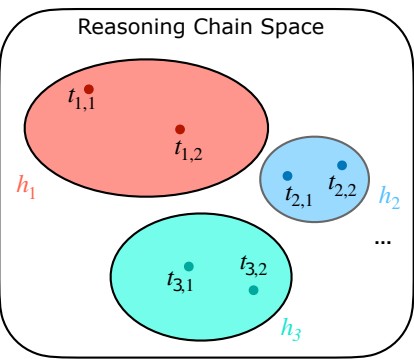

Figure 3: Illustration of the partitioning of the reasoning chain space based on the high-level tactics and hints employed in the solution.

tic. Instead, our leader-follower strategy takes two steps to generate the low-level reasoning chain, which can be formulated as below using the marginal probability formula:

$$\Pr(A|Q) = \sum_h \Pr(A|h, Q) \cdot \text{Unif}(h \in H) \cdot \Pr(H|Q) \tag{1}$$

Here, $\Pr(A|h, Q)$ corresponds to $\pi_{low}$, $\Pr(H|Q)$ corresponds to $\pi_{high}$, and $\text{Unif}(h \in H)$ denotes a unform distribution among $h \in H$. Note that $\text{Unif}(h \in H) \cdot \Pr(H|Q) \neq \Pr(h|Q)$. To illustrate with Fig. 3, suppose $H = \{h_1, h_2, h_3\}$ and $\Pr(H; \pi_{high}) = 1$, then sampling by $\Pr(h_i|Q)$ corresponds to sampling by the area of $h_i$, whereas sampling by $\text{Unif}(h \in H) \cdot \Pr(H|Q)$ corresponds to uniformly sampling among $h_1$, $h_2$, and $h_3$, i.e., regarding them as if they were having the same area. **For general cases, our strategy samples all the different hints returned by $\pi_{high}$ with equal probabilities.** Our strategy, therefore, aligns with the spirit of common practice in reinforcement learning to encourage the exploration behavior by making the density of actions more uniform (e.g., $\epsilon$-greedy policy reduces the chance of the best action and increases the chance of worse actions).

**Effectively and efficiently selecting final reasoning chains.** Given $n$ high-level tactics and hints $\{h_i\}_{i=1}^n$ proposed by the leader $\pi_{high}$, the low-level follower policy $\pi_{low}$ produces $n$ groups of reasoning chains $\{T_i\}_{i=1}^n$. Throughout our experiments, we maintain a constant size of reasoning chains for all groups, i.e., $\forall i, |T_i| = m$. As it is a common scenario that not all of the suggested tactics are relevant to problem solving, and irrelevant hints could make the low-level policy more susceptible to reasoning mistakes, we would like to *effectively* select the desired reasoning chains among those generated by the low-level policy. We would also like to make the selection process *efficient*, reducing the need to invoke a large number of language model calls.

To this end, we propose the following **tournament-based approach** to select reasoning chains: Within each group of reasoning chains $T_i$, we conduct majority voting to establish the group-majority answer $A_i$. Then, for every group, we randomly select a *single* reasoning chain from those that reach the group-majority answer, and we add it to a "selection" set $S$ (as a result, $|S| = n$). Next, denote the "final" reasoning chain as $\tau_{\text{final}}$, which we initialize as first reasoning chain of $S$. We then initiate a "tournament": Over $n - 1$ iterations, for each iteration $i$, we prompt GPT-4 to compare the current $\tau_{\text{final}}$ with the $(i + 1)$-th reasoning chain in $S$ to determine which is better. If the latter is better, we set it as $\tau_{\text{final}}$. Empirically, we also gather comparison results through majority voting conducted over $k$ repetitions.

The above approach requires a small number of $n \times k$ language model calls to select a desired reasoning chain. For instance, when we have $n = 4$ reasoning groups, each containing $m = 16$ reasoning chains, and we perform $k = 3$ comparison repetitions, it takes only 12 calls to GPT-4. Later, we will also demonstrate the effectiveness of our reasoning chain selection approach in improving final answer accuracy.

## 4  EXPERIMENTS

In this section, we perform quantitative and qualitative evaluations to demonstrate the effectiveness of our approach. We first investigate whether our approach successfully enhances the discovery and visibility of correct solutions, introducing a quantitative metric for this assessment. Subsequently, we assess whether our approach improves final answer accuracy in challenging mathematical reasoning problems. Lastly, we further analyze the success and failures our approach, as well as discuss its limitations.

**Experiment setup.** Unless otherwise specified, we adopt the MATH dataset (Hendrycks et al., 2020) and use its **Level-5** test set, i.e., the hardest subset of questions, to evaluate different approaches. For the high-level policy $\pi_{high}$, we adopt the two approaches outlined in Sec. 3: (1) prompt GPT-4 to generate at most $n$ relevant hints and tactics for a target question; (2) use SentenceBERT to retrieve $n$ most relevant problems from the MATH training set. For the low-level policy $\pi_{low}$, we adopt either GPT-3.5 or GPT-4. Unless otherwise specified, we establish the following parameters: $n = 4$ high-level hints (or reasoning groups) per question, $m = 16$ generated reasoning chains per group, and $k = 1$ comparison repeats for our tournament-based reasoning chain selection process (which is performed using GPT-4). We set temperature to be 0.3 during reasoning chain selection and 0.7 otherwise.

Our evaluation set of questions is constructed as follows: For experiments in Sec. 4.1 and Sec. 4.2, we randomly sample 20 questions for each of the 7 categories in the MATH Level-5 test set, resulting in an evaluation set of 140 questions. We opted for this smaller evaluation set due to the cost associated with evaluating our approach when either GPT-3.5 or GPT-4 serves as $\pi_{low}$, which amounted to approximately \$150 for these 140 questions. Evaluating GPT-4 as $\pi_{low}$ on the full test set would have been expensive. Later, when we conduct ablations in Sec. 4.3, we employ the entire subset of MATH Level-5 questions (excluding those requiring visual understanding in vector graphics) to assess our approach when GPT-3.5 serves as $\pi_{low}$. The findings remain consistent.

### 4.1  DO WE ENHANCE THE DISCOVERY AND VISIBILITY OF CORRECT SOLUTIONS?

In this section, we investigate whether by framing LLMs as a hierarchical policy and encouraging them to explore multiple diverse problem-solving tactics and hints, we improve the discovery and visibility of desired solutions that reach the correct answers. We compare our approach with the Chain-of-Thought (CoT) Sampling (Wei et al., 2022) + Majority Voting (Wang et al., 2022) baseline, which samples the same amount of detailed reasoning chains as ours and performs majority voting to obtain the final answer.

**We would like to assess the "visibility" of the correct answers among solutions generated along the exploration process.** *A correct answer is "visible" if it not only exists in at least one of reasoning chains, but also occupies a substantial proportion of them, even though it might not be the majority answer.* An intuitive way of measurement is to compare the accuracy and recall of correct answers between our approach and the baseline. **However, we find that standard metrics like accuracy and recall do not perfectly align with our goal.** In particular, for our CoT Sampling

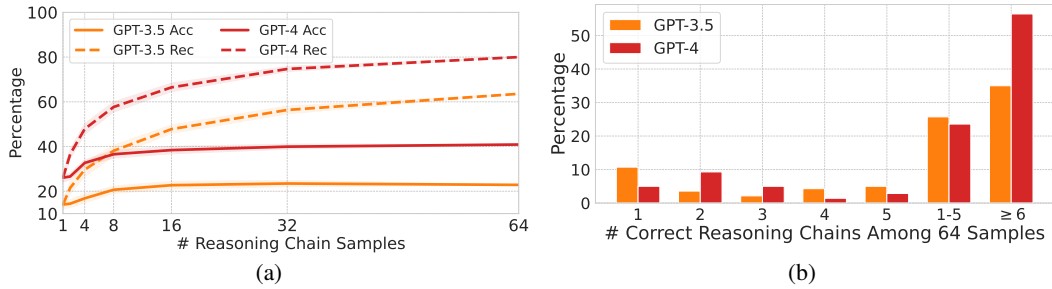

(a)          (b)

Figure 4: Statistics for the "CoT Sampling + Majority Voting" baseline. **(a)** Recall of correct answer steadily improves as the number of sampled reasoning chains increases, but the final answer accuracy after majority voting plateaus after a few reasoning chain samples. This suggests that the recall metric poorly correlates with the prominence of correct answers. **(b)** Percentage of questions where $x$ reasoning chains have correct answers among the 64 reasoning chain samples ($x \in \mathbb{N}$). It is a common scenario that even if LLM identifies correct solutions by a reasonable chance (and does so multiple times), these correct solutions are not effectively discovered and remain hidden during majority voting. *Such scenarios are not reflected in the standard accuracy metric. Therefore, standard accuracy and recall are not ideal metrics to assess the visibility of correct answers.*

| Model | Method | Alg. | Count. & Prob. | Geom. | Interm. Alg. | Num. Theory | Prealg. | Precal. | Overall |
|-------|--------|------|------|------|------|------|------|------|------|
| GPT-3.5 | CoT & Sampling | 46.67 | **32.83** | 11.67 | **17.50** | 29.84 | 69.58 | **10.31** | 31.19 |
| | Ours - Prompt for Tactics | 41.67 | **32.83** | 19.38 | 15.00 | 35.00 | **75.00** | 10.00 | 32.69 |
| | Ours - Retrieval | **49.48** | 31.98 | 15.31 | 15.00 | **45.31** | 72.50 | 10.00 | **34.23** |
| GPT-4 | CoT & Sampling | 62.50 | **55.67** | 50.00 | 24.83 | 61.67 | 88.75 | 12.00 | 50.78 |
| | Ours - Prompt for Tactics | 65.00 | 52.83 | 51.25 | 29.31 | **79.08** | 91.25 | **12.50** | **54.46** |
| | Ours - Retrieval | **67.50** | 44.58 | **53.50** | **31.00** | 60.17 | **95.00** | 7.83 | 51.36 |

Table 2: Comparison of the Grouped-Majority Recall metric between the Chain-of-Thought sampling baseline and our two exploration approaches outlined in Sec. 3. We use GPT-3.5 or GPT-4 as the language model for the CoT & Sampling Baseline, along with the low-level follower policy in our approaches. Metrics are obtained using $n = 4$ reasoning groups and $m = 16$ reasoning chains per group on our 140-question MATH Level-5 evaluation set.

baseline, as the number of reasoning chain samples goes up, the recall steadily improves, but the final-answer accuracy after majority voting plateaus after a few reasoning chain samples, as illustrated in Fig. 4a. This suggests that the standard recall metric poorly correlates the prominence of correct answers. On the other hand, the standard accuracy metric does not reflect the scenarios where LLM identifies correct solutions by a reasonable chance (and does so multiple times), but these correct answers become submerged during majority voting. Such scenarios often occur, and we illustrate this phenomenon in Fig. 4b. Therefore, the visibility of desired solutions is not adequately captured by the standard accuracy or recall metric.

We invent a "microscope" to inspect the visiblity of the correct answers. **We propose the following *"Grouped-Majority Recall"* metric to better quantify the visibility of correct solutions.** The calculation takes two steps. First of all, recall that our proposed method produces $n$ groups each containing $m$ reasoning chains. The CoT Sampling baseline can be viewed as a special case of our method with a single group ($n = 1$) of empty high-level tactics. To calculate the Grouped-Majority Recall metric, we first perform majority voting in each group to obtain $n$ majority answers, and then we calculate the percentage of questions whose ground truth answer exist in *at least* one of the $n$ group-majority answers. Sometimes, a group will contain multiple majority answers, in which case we calculate the expected value of our metric. To compare our approach with the CoT Sampling baseline, we randomly partition $n \times m$ CoT samples into $n$ groups and calculate our metric in the same manner.

| Model | Method | Alg. | Count. & Prob. | Geom. | Interm. Alg. | Num. Theory | Prealg. | Precal. | Overall |
|---|---|---|---|---|---|---|---|---|---|
| GPT-3.5 | CoT Sampling + Voting | 30.00 | 25.00 | 10.00 | 15.00 | 5.00 | 60.00 | **10.00** | 22.14 |
| | ToT + Voting | 15.00 | 0.00 | 5.00 | **20.00** | 0.00 | 49.17 | 0.00 | 13.45 |
| | Ours - Prompt for Tactics | 40.00 | **30.00** | **15.00** | 0.00 | 20.00 | 65.00 | 5.00 | 25.00 |
| | Ours - Retrieval | **50.00** | 25.00 | **15.00** | 0.00 | **30.00** | **70.00** | 5.00 | **27.85** |
| GPT-4 | CoT Sampling + Voting | 50.00 | 36.25 | **47.50** | 10.00 | 55.00 | 75.00 | **12.50** | 40.89 |
| | Ours - Prompt for Tactics | **65.00** | 35.00 | 40.00 | 15.00 | **70.00** | **80.00** | 5.00 | **44.29** |
| | Ours - Retrieval | 50.00 | **45.00** | 40.00 | **30.00** | 50.00 | 70.00 | 10.00 | 42.14 |

Table 3: Comparison of final answer accuracy between the Chain-of-Thought sampling baseline and our two exploration approaches outlined in Sec. 3. We also report Tree-of-Thoughts results for reference. We use GPT-3.5 or GPT-4 as the language model for CoT & ToT, along with the low-level policy in our approaches. Results are reported on our 140-question MATH Level-5 evaluation set.

*Intuition behind the Grouped-Majority Recall:* In contrast to the standard accuracy metric, rare correct answers that might be obscured amidst all the $n \cdot m$ samples could emerge as the majority in one of the groups, thereby being recognized by our new metric. This occurrence, as noted in our observations, is not uncommon: when a high-level tactic is accurate yet seldom sampled, it frequently yields a substantial number of correct low-level solutions that can dominate in a group, despite remaining a minority among all samples. Our new metric aptly acknowledges such instances. Additionally, *in contrast to the standard recall metric*, for a correct answer to be recognized by our new metric, it should not only appear in at least one reasoning chain, but also take up the majority in a group, necessitating its appearance in multiple reasoning chains.

**Results.** We report the Grouped-Majority Recall metrics in Tab. 2. We find that our exploration approaches effectively enhance Grouped-Majority Recall when either GPT-3.5 or GPT-4 serves as our low-level policy, demonstrating that our methods improve the discovery and visibility of solutions leading to the correct answers. Additionally, we observe that among our two exploration approaches, using concise technique-based hints underperforms using retrieved problem-solution hints when GPT-3.5 is used as the follower, but outperforms the latter for GPT-4. We conjecture that this is caused by the stronger ability for the GPT-4 follower to understand and effectively utilize the hints in specific problem-solving processes. On the other hand, for the weaker GPT-3.5 follower, even if the high-level hints are already inspiring and meaningful, it may not effectively utilize the hint to solve the target problem. An example is illustrated in Tab. 9.

## 4.2 Do We Improve Final Answer Accuracy for Challenging Math Problems?

Next, we investigate whether our exploration and tournament-based reasoning chain selection approach enhance the final answer accuracy on the Level-5 test set of the MATH dataset. We compare our approach with the CoT Sampling + Majority Voting baseline in Tab. 3. We find that both of our exploration approaches successfully improve the final answer accuracy, demonstrating that our approach effectively selects among the explored reasoning chains to retain the high-quality ones.

We also implement Tree-of-Thoughts (ToT) (Yao et al., 2023) for mathematics problem solving following the original paper, and present its results on GPT-3.5 as a reference[1]. We perform breadth-first search (BFS) in ToT, where at each step we expand 8 children and keep the best 5 candidates at each depth level. We limit the tree depth to 16. However, we observe that the final accuracy for ToT is significantly worse than the CoT Sampling + Voting baseline. Upon further analysis, we find that the average number of reasoning chains ToT produces is 8.41, which is significantly fewer than the 64 reasoning chains in our baseline, potentially harming its performance.

---

[1]Running ToT is especially expensive and costs over $100 on GPT-3.5. Running ToT on GPT-4 would incur thousands of dollars of expense, so we would like to leave it for future work.

| Method | Alg. | Count. & Prob. | Geom. | Interm. Alg. | Num. Theory | Prealg. | Precal. | Overall |
|---|---|---|---|---|---|---|---|---|
| CoT Sampling + Voting | 65.84 | 39.97 | 19.33 | 11.06 | 38.75 | 69.11 | 4.74 | 40.63 |
| Ours - Prompt for Tactics | 64.77 | **42.84** | **26.00** | 12.23 | 42.47 | 71.91 | **11.40** | 42.58 |
| Ours - Retrieval | **69.81** | 40.00 | 22.25 | **15.26** | **43.43** | **72.65** | 8.87 | **44.29** |

(a) Grouped-Majority Recall.

| Method | Alg. | Count. & Prob. | Geom. | Interm. Alg. | Num. Theory | Prealg. | Precal. | Overall |
|---|---|---|---|---|---|---|---|---|
| CoT Sampling + Voting | 56.73 | 26.57 | 16.00 | 6.56 | 23.27 | 60.74 | 4.49 | 32.22 |
| Ours - Prompt for Tactics | 55.00 | **33.96** | **24.00** | 8.61 | **32.47** | 63.76 | **6.74** | 35.15 |
| Ours - Retrieval | **57.86** | 29.25 | 20.00 | **11.07** | 31.82 | **66.44** | 5.62 | **36.10** |

(b) Final answer accuracy.

Table 4: Comparison of Grouped-Majority Recall and final answer accuracy on our 1047-question evaluation set using GPT-3.5 as the language model for the CoT Sampling Baseline along with the low-level follower policy in our approaches.

### 4.3 MORE RESULTS AND ABLATION STUDY

**Evaluation on a larger set of MATH questions**. As stated in our experiment setup, our experiments in Sec. 4.1 and Sec. 4.2 were conducted using a smaller sample of 140 questions from the MATH Level-5 test-set due to the cost associated with evaluating GPT-4. In this section, we expand our evaluation by using GPT-3.5 as the low-level follower $\pi_{low}$ and evaluating on a larger set of 1047 questions from the MATH Level-5 test set. This evaluation set encompasses all questions from the MATH Level-5 test set, except those that include Asymptote (a Vector Graphics Language) code in the question or answer, as these questions require visual comprehension. Additionally, unlike the 140-question evaluation set, which features an equal distribution of 20 questions from each of the 7 categories, the 1047 questions evaluated in this section exhibit an uneven distribution among different categories, with 429 questions coming from algebra and prealgebra, which are considered easier compared to the rest.

We present the Grouped-Majority Recall evaluation results in Tab. 4a and the final answer accuracy results in Tab. 4b. The findings are consistent with those we obtained in Sec. 4.1 and Sec. 4.2. We also observe that the overall Grouped-Majority Recall and the final answer accuracy are higher than those we obtained on the 140-question evaluation set, which is due to the higher portion of algebra and prealgebra questions in our 1047-question evaluation set, and these questions are considered easier.

**Ablation on tournament-based reasoning chain selection**. Additionally, we perform an ablation on different design decisions of our tournament-based reasoning chain selection approach. We conduct an experiment where we use either GPT-3.5 or GPT-4 to perform our tournament selection process, while varying the value of $k$, the number of comparison repetitions, in each of iteration of the tournament. Results are shown in Tab. 5. We find that when $k$ is small, using GPT-4 for tournament-based reasoning chain selection leads to a significant improvement compared to using GPT-3.5. Notably, GPT-4 demonstrates strong performance as a reasoning chain selector even with $k = 1$, resulting in a noteworthy enhancement of final answer accuracy over the CoT Sampling baseline.

We also conduct an experiment where we vary the decoding temperature ($T$) of GPT-4 during our tournament-based reasoning chain selection process (the temperature was set to 0.3 in our previous experiments). Results are shown in Tab. 6. We observe that when $0 < T < 1$, different temperatures have little effect on the final answer accuracy.

**Does final answer accuracy improvement come from our hierarchical policy framework and / or our tournament selection process?** We perform an ablation experiment where we investigate

| Method | Ours - Retrieval | | | Ours - Retrieval | | | CoT Sampling + Voting |
|---|---|---|---|---|---|---|---|
| Tournament Model | GPT-3.5 | | | GPT-4 | | | N/A |
| # Comparisons | $k = 1$ | $k = 3$ | $k = 5$ | $k = 1$ | $k = 3$ | $k = 5$ | N/A |
| Answer Accuracy | 21.43 | 22.86 | **27.14** | **27.85** | **27.85** | **27.85** | 22.14 |

Table 5: Ablation on using different models to conduct our tournament-based reasoning chain selection process, along with using different $k$, i.e., different numbers of comparison repetitions, during this process. We compare our retrieval-based method with the CoT Sampling baseline. Results are obtained using our 140-question evaluation set. We use GPT-3.5 as the language model for our low-level follower and the CoT Sampling Baseline.

| Temperature | $T = 0$ | $T = 0.3$ | $T = 0.7$ | $T = 1.0$ |
|---|---|---|---|---|
| Answer Accuracy | 27.14 | **27.85** | **27.85** | **27.85** |

Table 6: Ablation on using different temperature ($T$) in our tournament-based reasoning chain selection process. Results are obtained using GPT-4 as our tournament selection model with $k = 1$ comparison repetition on our 140-question evaluation set. We use GPT-3.5 as the language model for our low-level follower and the CoT Sampling Baseline.

the role of our hierarchical policy framework and our tournament-based reasoning chain selection process in improving the final answer accuracy. Results are shown in Tab. 7. We find that

- For our hierarchical policy approaches, adopting our tournament-based reasoning chain selection process yields better final answer accuracy than using majority voting, demonstrating the effectiveness of our tournament selection process. Intuitively, this is because not all of the tactics and hints produced by our high-level leader are helpful, and some of them might mislead the low-level follower, potentially causing it to generate a consistent but wrong answer under a misleading high-level guidance. By evaluating reasoning chains using our tournament-based selection approach, we effectively remove those that exhibit reasoning mistakes and keep those that are more trustful.

- Additionally, for the CoT + Sampling baseline, adopting our tournament-based reasoning chain selection process does not ourperform our approaches that employ the hierarchical-policy framework. This demonstrates that our hierarchical policy plays a significant role in enhancing LLM's ability to solve challenging problems.

**Evaluation on GSM8K**. Previously, our experiments were conducted on the Level-5 subset of the MATH dataset. In this experiment, we investigate the efficacy of our approach on a wider range of datasets such as GSM8K. Results are shown in Tab. 8. We find that while GSM8k features easier mathematics problems than the MATH Level-5 questions used in our previous experiments, our approach continues to achieve better final-answer accuracy. This suggests that our approach retains its effectiveness across datasets with varying degrees of difficulty.

## 4.4 FURTHER ANALYSIS AND LIMITATIONS

In this section, we provide a further analysis into the success and failures of our approach, and we discuss the potential limitations of our approach along this process.

We observe that our high-level leader policy is capable of producing many insightful and inspiring hints, such as the "High-Level Thought 1" in Fig. 2, even though it sometimes produces irrelevant hints (e.g., "High-Level Thought 2" in Fig. 2). To further analyze the quality of generated high-level hints, we perform an experiment, where we aggregate all the high-level techniques and concepts produced by our first hint generation approach into a set. For each hint, we obtain its corresponding problem and ground truth answer, and then prompt GPT-4 to generate 10 "ground-truth hints" given both information. Subsequently, we calculate the percentage of hints that match at least one of its corresponding "ground-truth hints", serving as a measure of hint quality. Our findings reveal that

| Method | Baseline CoT + Sampling | Ours - Tactic | Ours - Retrieval |
|---|---|---|---|
| Majority Voting | 22.14 | 21.79 | 23.57 |
| Majority Voting over Groups | 19.70 | 20.24 | 23.39 |
| Tournament | 22.86 | **25.00** | **27.85** |

Table 7: Effect of majority voting and our tournament-based reasoning chain selection on the final-answer accuracy of the CoT + Sampling baseline and our hierarchical policy approaches. For "Majority Voting", we directly perform majority voting over the 64 sampled reasoning chains per problem. For "Majority Voting over Groups" and "Tournament", we adopt $n = 4$ groups, each having $m = 16$ reasoning chains. We use GPT-3.5 as the language model for our low-level follower and the CoT Sampling Baseline.

| Method | CoT + Sampling Baseline Majority Voting | CoT + Sampling Baseline Tournament | Ours - Retrieval (w/ Tournament) |
|---|---|---|---|
| Answer Accuracy | 88.45 | 88.85 | **89.91** |

Table 8: Final answer accuracy on GSM8K. We adopt GPT-3.5 as the low-level follower, and we sample 32 reasoning chains per problem. For "Majority Voting", we directly perform majority final-answer voting over the 32 reasoning chains. For tournament, we adopt $n = 4$ groups, each having $m = 8$ reasoning chains.

**94.3%** of hints are among the ground-truth hints, highlighting the effectiveness of our approach in proposing pertinent, insightful, and inspiring hints.

Next, we delve into the analysis of common sources of failure cases to gain a deeper understanding of our approach's behavior. One significant factor contributing to these failures is that, despite the high-level leader producing hints being both inspiring and meaningful, the low-level follower may not closely follow these hints to solve the target problem, resulting in reasoning errors. Sometimes, it might even ignore the hints. An illustrative example is presented in Tab. 9. We also empirically observe that such phenomenon occurs more often for a weaker follower language model compared to a stronger follower language model (e.g., GPT-3.5 vs. GPT-4). Furthermore, even when the follower effectively incorporates the hints into its problem-solving processes, reasoning errors can still occur, as illustrated in Tab. 10. The inconsistent adherence to high-level hints and the dependence on the capabilities of the follower model highlight the current limitations of our approach.

## 5 CONCLUSION

In this work, we propose to frame large language models (LLMs) as a hierarchical policy to effectively explore the expansive solution spaces in challenging mathematical reasoning problems. Our hierarchical policy framework consists of a visionary "high-level" leader policy, which establishes connections between the target problem and the language model knowledge to generate hints, along with a "low-level" follower policy that leverages these hints as an in-context guidance throughout detailed problem-solving processes. Within this framework, we introduce two effective approach for the high-level leader policy to generate a diverse set of problem-solving tactics and hints for exploration. Additionally, we present an effective and efficient tournament-based method to select desired reasoning chains among the explored ones to attain the final answer. Through experiments, we demonstrate that our approach enhances problem-solving strategy exploration, improves the discovery and visibility of correct solutions, and enhances the final answer accuracy on challenging problems in the MATH dataset.

---

**Question:** Carson flips over the cards of a standard 52-card deck one at a time. What is the probability that he flips over the ace of spades before any face card (jack, queen or king)?

**High-Level Hint:** Conditional Probability

**Guidance:** Use principles of "Conditional Probability" to approach this question. Ensure you think step-by-step, presenting a clear, logical solution pathway. Conclude with the final answer enclosed in the \boxed command.

---

**GPT-3.5 Follower Output:**

To solve this problem, we can consider the probability of flipping over the ace of spades first and the probability of flipping over any face card first, and then subtract the probability of flipping over any face card first from 1. There is only one ace of spades in the deck, so the probability of flipping it over first is $\frac{1}{52}$. There are 12 face cards (four jacks, four queens, and four kings) in the deck. So the probability of flipping over any face card first is $\frac{12}{52}$. Therefore, the probability of flipping over the ace of spades before any face card is $1 - \frac{12}{52} = \boxed{\frac{5}{13}}$.

---

Table 9: A failure case in which the GPT-4 high-level leader generates a highly-relevant hint for the question, but the GPT-3.5 low-level follower fails to utilize the hint effectively during problem-solving, resulting in wrong answer.

---

**Retrieved Question:** For some constants $a$ and $b$, let $f(x) = \begin{cases} ax + b & \text{if } x < 2 \\ 8 - 3x & \text{if } x \geq 2 \end{cases}$

The function $f$ has the property that $f(f(x)) = x$ for all $x$. What is $a + b$?

**Answer:**

Setting $x = 3$, we get $f(3) = -1$. Since $-1 < 2$, $f(-1) = -a + b$.
Hence, $f(f(3)) = f(-1) = -a + b$. But $f(f(x)) = x$ for all $x$, so $-a + b = 3$.
Setting $x = 4$, we get $f(4) = -4$. Since $-4 < 2$, $f(-4) = -4a + b$.
Hence, $f(f(4)) = f(-4) = -4a + b$. But $f(f(x)) = x$ for all $x$, so $-4a + b = 4$.
Subtracting the equations $-a + b = 3$ and $-4a + b = 4$, we get $3a = -1$, so $a = -1/3$.

From $-a + b = 3$, we get $b = a + 3 = 8/3$. Hence, $a + b = (-1/3) + 8/3 = \boxed{\frac{7}{3}}$.

---

**Question:** For some constants $a$ and $b$, let $f(x) = \begin{cases} 9 - 2x & \text{if } x \leq 3 \\ ax + b & \text{if } x > 3 \end{cases}$

The function $f$ has the property that $f(f(x)) = x$ for all $x$. What is $a + b$?

---

**GPT-4 Follower Output:**

Setting $x = 4$, we get $f(4) = 4a + b$. Since $4a + b > 3$, $f(4a + b) = 9 - 2(4a + b)$.
Hence, $f(f(4)) = f(4a + b) = 9 - 2(4a + b)$. But $f(f(x)) = x$ for all $x$, so $9 - 2(4a + b) = 4$.
Setting $x = 2$, we get $f(2) = 5$. Since $5 > 3$, $f(5) = 5a + b$.
Hence, $f(f(2)) = f(5) = 5a + b$. But $f(f(x)) = x$ for all $x$, so $5a + b = 2$.
Subtracting the equations $9 - 2(4a + b) = 4$ and $5a + b = 2$, we get $3a + 3 = 2$, so $a = -1/3$.

From $5a + b = 2$, we get $b = 2 - 5a = 5/3$. Hence, $a + b = (-1/3) + 5/3 = \boxed{\frac{4}{3}}$.

---

Table 10: A failure case in which a very similar question along with its ground truth solution (from the MATH training set) are retrieved and padded as prompt for the GPT-4 low-level follower. Even though the model effectively incorporates the provided demonstration into its problem-solving process by following similar reasoning steps, the model still fails to produce the correct answer.

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

## A    ANSWER EXTRACTION

We obtain the final answer from a generated reasoning chain by identifying the content within the LaTeX environment \boxed{}, as specified in the prompt shown in Tab. 13.

## B    GPT 3.5/4 MODEL VERSION AND HYPERPARAMETER DETAILS

We use "gpt-3.5-turbo-0613'' and "gpt-4-0613" as the GPT-3.5 and GPT-4 version used throughout the experiments in our paper. We set the decoding temperature to 0.3 for tournament-based reasoning chain selection and 0.7 otherwise (i.e., for hint generation from the high-level leader, reasoning chain generation from the low-level follower, along with the baselines).

## C    COST ANALYSIS

| Method
Low-Level Follower | CoT Baseline
GPT-3.5 | Ours - Tactics
GPT-3.5 | Ours - Retrieval
GPT-3.5 | CoT Baseline
GPT-4 | Ours - Tactics
GPT-4 | Ours - Retrieval
GPT-4 |
|---|---|---|---|---|---|---|
| Sampling | 10.72 | 11.71 | 9.83 | 204.16 | 191.11 | 188.47 |
| Tournament | None | 2.42 | 4.45 | None | 4.52 | 5.92 |
| Total | 10.72 | 14.12 | 14.28 | 204.16 | 195.63 | 194.39 |

Table 11: Cost comparison between our approach and the CoT + Sampling Baseline on our 140-question MATH Level-5 evaluation set. We generate $n \times m = 4 \times 16 = 64$ reasoning chains per question. GPT-4 is utilized to perform tournament selection in our approaches.

| Method
Low-Level Follower | CoT Baseline
GPT-3.5 | Ours - Tactics
GPT-3.5 | Ours - Retrieval
GPT-3.5 | CoT Baseline
GPT-4 | Ours - Tactics
GPT-4 | Ours - Retrieval
GPT-4 |
|---|---|---|---|---|---|---|
| Avg. # Input Tokens Per Question | 0.11K | 0.73K | 1.88K | 0.11K | 0.88K | 2.00K |
| Avg. # Output Tokens Per Question | 38.20K | 41.61K | 34.32K | 24.25K | 22.85K | 22.14K |

Table 12: Comparison on the number of input and output tokens per-question between our approach and the CoT + Sampling Baseline on our 140-question MATH Level-5 evaluation set. We generate $n \times m = 4 \times 16 = 64$ reasoning chains per question. GPT-4 is utilized to perform tournament selection in our approaches. Tournament tokens are included in the table.

## D    PROMPTS

### D.1    CHAIN-OF-THOUGHT (COT) BASELINE PROMPT

**Question:**
{question}

Please provide step-by-step reasoning, and present the final answer in the LaTeX environment starting with \boxed{.

**Answer:**

Table 13: Zero-shot prompt for reasoning chain generation using the baseline CoT Sampling + Majority Voting approach on the MATH dataset. We need the model to present the final answer within a LaTeX **boxed** environment, which is capable of effectively handling complex output formats, such as intervals or matrices.

### D.2    PROMPTS FOR LOW-LEVEL FOLLOWER

---

**Question**:
{question}

Use the methods related to "{technique or concept}" to derive your answer!
Detail your reasoning step-by-step.
Finally, present your final answer using the LaTeX command "\boxed{".

**Answer:**

---

Table 14: Zero-shot prompt for following a concise technique and concept produced by the high-level leader policy.

---

**Question**:
{demo-question}

**Answer**:
{demo-step-by-step-answer}

Please refer to the above example as a demonstration. If it is not relevant to the current question, you may disregard them.

**Question**:
{question}
**Answer**:
Please provide step-by-step reasoning, and present the final answer in the latex environment starting with \boxed{, without using a diagram.

---

Table 15: Zero-shot prompt for following a question-answer demonstration retrieved by the high-level leader policy.

### D.3 PROMPTS FOR HIGH-LEVEL LEADER TO GENERATE TACTICS AND HINTS

---

When given a mathematical problem, your task is to **list high-level mathematical techniques or concepts** that can potentially lead to its solution. Do not provide detailed explanations, only name the techniques.Each technique you list should have the potential to guide towards a solution on its own. For clarity, here are some examples:

**Example 1**:
**Question**:
Trapezoid $ABCD$ has sides $AB = 92$, $BC = 50$, $CD = 19$, and $AD = 70$, with $AB$ parallel to $CD$. A circle with center $P$ on $AB$ is drawn tangent to $BC$ and $AD$. Given that $AP = \frac{m}{n}$, where $m$ and $n$ are relatively prime positive integers, find $m + n$.

**Response**:
1: Angle Bisector Theorem
2: Power of a Point Theorem
3: Similar Triangles

**Example 2**:
**Question**:
Let $P$ be a point on the line $\begin{pmatrix} 3 \\ -1 \\ 2 \end{pmatrix} + t \begin{pmatrix} 2 \\ -2 \\ 1 \end{pmatrix}$ and let $Q$ be a point on the line $\begin{pmatrix} 0 \\ 0 \\ 4 \end{pmatrix} + s \begin{pmatrix} 1 \\ 2 \\ -1 \end{pmatrix}$. Find the shortest possible distance $PQ$.

**Response**:
1: Vector Geometry and Line Equations
2: Parametric Equations for Lines in 3D
3: Distance formula in 3D space
4: Minimization with Derivatives

**Example 3**:
**Question**:
Twenty switches in an office computer network are to be connected so that each switch has a direct connection to exactly three other switches. How many connections will be necessary?

**Response**:
1: Handshaking Lemma in Graph Theory
2: Combinatorial counting principles

**Example 4**:
**Question**:
What is the sum of the $x$-values that satisfy the equation $5 = \frac{x^3 - 2x^2 - 8x}{x+2}$?

**Response**:
1: Factorization Method
2: Rational Root Theorem
3: Vieta's Formula
4: Polynomial Long Division

**Example 5**:
**Question**:
{question}

**Response**:

---

Table 16: Few-shot prompt for the high-level leader policy to propose diverse techniques and concepts that serve as hints to steer the problem-solving process of the low-level follower.

### D.4 PROMPT FOR TOURNAMENT-BASED REASONING CHAIN SELECTION

---

**Question**:
{question}

Presented below are two possible answers:

**Answer 1**:
{reasoning-chain-1}

**Answer 2**:
{reasoning-chain-2}

Examine the difference between two answers thoroughly. Which one do you consider to be the most accurate? Support your decision with a comprehensive explanation. Provide your preference by stating with "Preferred Answer Index:" and the best answer.

---

Table 17: Zero-shot prompt for comparing two reasoning chains in our tournament-based reasoning chain selection process.

D.5 PROMPT FOR TREE-OF-THOUGHTS (ToT)

**Answering Guidelines**
1. Please answer the question step by step.
2. Organize each step into **two lines**:
- The first line starts with "#" followed by the step number and then succinctly describe the key action or result of the step.
- The second line begins with "-" and provides a brief reasoning or explanation for the particular step.
3. The first line of the final step is "What is the final answer?", and the second line provides the final answer using the latex format \boxed{} in this step.

**Example**
**Question**:
By starting with a million and alternatively dividing by 2 and multiplying by 5, Anisha created a sequence of integers that starts 1000000, 500000, 2500000, 1250000, and so on. What is the last integer in her sequence? Express your answer in the form $a^b$, where $a$ and $b$ are positive integers and $a$ is as small as possible.
**Answer**:
Let's think step by step!
#1. How can we express a million in terms of its prime factors?
- A million can be expressed as $10^6$. Further breaking it down gives: $10^6 = (2^6)(5^6)$.

#2. What happens to the sequence as we proceed?
- Each time she divides by 2, she removes one factor of 2. Each time she multiplies by 5, she adds one factor of 5. Since she starts with 6 factors of 2, after 6 divisions by 2, all factors of 2 are removed.

#3. How many factors of 5 does she have at the end of the process?
- Initially, there are $5^6$. After 6 steps of multiplication by 5, 6 more factors of 5 are added. This gives a total of $5^6 \times 5^6 = 5^12$.

#4. What is the last integer in her sequence?
- At the end of 12 steps, every factor of 2 has been replaced with a factor of 5. Thus, the integer becomes $5^6 \times 5^6 = 5^{12}$.

#5. What is the final answer?
- $\boxed{5^{12}}$.

Please follow the **Answering Guidelines** and use the format shown in the above example to answer the following question!!

**Question**:
{question}
**Answer**:
Let's think step by step!
{current-steps}

Table 18: One-shot prompt for generating the next reasoning step in ToT.

**Instruction**: Given a problem statement and its current reasoning steps, determine if the proposed additional reasoning step is useful for solving the problem. Note: a useful reasoning step does not need to be directly related to the final question.

**Question**:
Let $x, y$, and $z$ be real numbers such that $x - 2y + 2z = 3$ and $2x + y - z = 6$. Find $x^2 + z^2 - y^2$.
**New Reasoning Step**:
#1. Solve the system of equations to find the values of x, y, and z.
**Is the new reasoning step necessary or useful?**
No. We can't find unique values for x, y, z with only two linear equations for three unknowns.

**Question**:

Let $P$ be a point on the line $\begin{pmatrix} 3 \\ -1 \\ 2 \end{pmatrix} + t \begin{pmatrix} 2 \\ -2 \\ 1 \end{pmatrix}$ and let $Q$ be a point on the line

$\begin{pmatrix} 0 \\ 0 \\ 4 \end{pmatrix} + s \begin{pmatrix} 1 \\ 2 \\ -1 \end{pmatrix}$. Find the shortest possible distance $PQ$.

#1. Find the vector connecting the initial points of the two lines, which corresponds to the direction from point $P$ to point $Q$.
**New Reasoning Step**:
#2. Calculate the dot product of the direction
vector of the first line with the direction vector of the second line.

**Is the new reasoning step necessary or useful?**
Yes. The dot product will help determine if the lines are parallel, which is important in calculating the shortest distance between them.

Evaluate the reasoning step in the new question by following the provided instruction, as shown in the previous examples.

**Question**:
{question}
{current-steps}
**New Reasoning Step**:
{new-reasoning-step}
**Is the new reasoning step necessary or useful?**

Table 19: Few-shot prompt for scoring the generated next reasoning step in ToT.

