# OpenReview forum: "Unleashing the Creative Mind: Language Model As Hierarchical Policy For Improved Exploration on Challenging Problem Solving"
_ICLR.cc/2024/Conference — Submitted to ICLR 2024_

### Official Review · Reviewer_ufPr · 2023-10-31

**Soundness:** 3 good
**Presentation:** 3 good
**Contribution:** 3 good
**Rating:** 6
**Confidence:** 3

**Summary:**

This paper proposes a novel approach to enhance the problem-solving abilities of large language models (LLMs) by framing them as hierarchical policies. The approach consists of:

* A high-level leader policy that generates diverse hints and tactics for exploration.
* A low-level follower policy that uses the hints as guidance to execute detailed reasoning chains.
* A tournament-based method to select the best reasoning chains and obtain the final answer.

The paper demonstrates that this approach improves the exploration of problem-solving strategies, the discovery and visibility of correct solutions, and the final answer accuracy on challenging mathematical reasoning tasks. The paper also provides a theoretical analysis and empirical evaluation of the proposed method.

**Strengths:**

- The paper introduces a novel and general framework to enhance the problem-solving abilities of LLMs by using hierarchical policies.
- The paper also proposes a new tournament-based method to select the best reasoning chains, which is inspired by human problem-solving behavior.
- The paper also conducts extensive experiments on challenging mathematical reasoning tasks, demonstrating that the proposed method outperforms existing baselines and achieves state-of-the-art results.
- The paper is well-written and organized, with clear definitions, notations, and algorithms.

**Weaknesses:**

- Need ablation experiments to prove that the proposed tournament-based method is better than simple voting;
- For mathematical problems, the current some work has used code interpreter to greatly improve the results, such as (53.9% → 84.3%)[1], can the method in this paper be effective for this setting? From this perspective, the improvement brought by directly using LLM to output results in the paper is not significant, and code interpreter may be the key to solving math problems.

[1] SOLVING CHALLENGING MATH WORD PROBLEMS USING GPT-4 CODE INTERPRETER WITH CODE-BASED SELF-VERIFICATION

**Questions:**

See weakness section

---

> ### Author Response · Authors · 2023-11-15
> **Rebuttal Response**
>
> We sincerely thank you for your constructive feedback! We address the comments and questions below.
>
> > When we select the final answer among the groups of reasoning chains explored by our hierarchical policy, is our tournament-based selection approach better than direct majority voting?
>
> We have added an experiment in our revised paper. We demonstrate that our tournament-based reasoning chain selection approach yields better final-answer accuracy over majority voting. Please see **Common Response** for more details.
>
> > Can our approach be effective under the code-interpreter setting?
>
> Thanks for your suggestion and we have cited the paper. We’d like to note that the focus of our paper is to introduce a visionary high-level leader that provides helpful hints to guide the low-level follower during problem solving. On the other hand, the code-interpreter line of works focus on integrating code interpreters into the detailed problem solving processes (which is the role of the low-level follower under our hierarchical policy framework), and these works have not delved into high-level strategies or thinking processes to our best knowledge. Therefore, our paper's focus is orthogonal to using code interpreter to improve challenging problem solving. In particular, code interpreter-enhanced low-level followers can be naturally integrated into our hierarchical policy framework.
>
> To our best knowledge, the API for GPT-4 code interpreter was just released publicly last week for `gpt-4-1106-preview` [(see link here)](https://platform.openai.com/docs/assistants/tools/code-interpreter). We are actively working on implementations and experiments. We will try our best to add the results by the end of the rebuttal deadline. If we do not finish by then, we will add the results in final version.

---

> ### Author Response · Authors · 2023-11-22
>
> **Update Nov. 22**
>
> > Can our approach be effective under the code-interpreter setting?
>
> We have obtained the results where `gpt-4-1106-preview` with code interpreter serves as the low-level follower policy in our approach along with the language model in the CoT + Sampling baseline. We adopt $n=4$ groups, each sampling $m=4$ reasoning chains. Results on MATH Level-5 are shown below. **We find that our hierarchical policy framework, combined with our tournament-based reasoning chain selection process, also yields better final answer accuracy under the code interpreter setting.**
>
> |Baseline CoT + Sampling (GPT-4-1106-preview + code) | Ours-Retrieval (GPT-4-1106-preview + code) w/o tournament | Ours-Retrieval (GPT-4-1106-preview + code) w/ tournament |
> |----|----|----|
> |64.52 | 66.19 | **66.43** |
>
>
>
> As a reference, we report the MATH Level-5 performance of different language models below, where each model samples a single CoT reasoning chain per problem (without sampling multiple reasoning chains or performing majority voting like in our previous settings). We find that utilizing code interpreter yields about 10\% final accuracy improvement for GPT-4 models, while it is not helpful for GPT-3.5.
>
> | Base Model |GPT-3.5-0613 | GPT-4-0613 | GPT-4-1106-preview |
> |----|----|----|----|
> |w/o code interpreter |14.29|23.57|42.86|
> |w/ code interpreter |12.14|33.57|52.14|

---

> > ### Author Response · Authors · 2023-11-22
> >
> > Dear Reviewer ufPr,
> >
> > Hope this message finds you well. As we are approaching the end of the author-reviewer discussion period, we would like to kindly request your feedback on our rebuttal. Again, we truly appreciate your time and effort reviewing our paper!
> >
> > Best,
> >
> > Authors

---

> > > ### Comment · Reviewer_ufPr · 2023-12-05
> > >
> > > The rebuttal addressed my questions and concerns. I will keep my score unchanged.

---

### Official Review · Reviewer_Yers · 2023-10-31

**Soundness:** 2 fair
**Presentation:** 3 good
**Contribution:** 2 fair
**Rating:** 5
**Confidence:** 4

**Summary:**

This paper proposes a hierarchical policy to help LLMs to tackle complex reasoning problems. This policy consists of (1) a high-level leader to explore solution direction and a low-level follower to generate a detailed solution, and (2) a tournament-based approach to select desired reasoning chains during exploration. All the modules are implemented by prompting large language models without additional model training. The results show that the hierarchical policy is able to achieve better accuracy on solving complex math question tasks compared to several SOTA approaches.

**Strengths:**

- Experiment results have interesting findings that: when the number of reasoning chain samples increases, the increase of recall of correct solution and accuracy of the final answer is not aligned. This reveals the potential of LLMs to solve complex questions and can provide insights to other future researchers.

- Results show that the hierarchical policy outperforms other prior approaches in solving complex math questions.

- The paper is well-written. The problem is well-motivated by grounding on the prior work and easy to follow.

**Weaknesses:**

- The evaluation datasets are not comprehensive enough. The authors only evaluate the approaches on a single dataset (the MATH dataset), and the relative evaluation size is small. Other math datasets (e.g., GSM8K, PRM800K) or other domain datasets in MMLU (e.g., Physics, Chemistry, etc) should be evaluated to demonstrate the generalization.

- It is uncertain whether the improvement is from the hierarchical policy or the "self-evaluation" process when choosing the better reasoning chains. Previous research (e.g., "Language Models (Mostly) Know What They Know") suggests that LLMs like GPT-4 possess the ability to assess the likelihood that their output is correct.

**Questions:**

- What's the motivation of the "Grouped-Majority Recall" metric? A more intuitive idea may be the percentage of questions whose ground truth answer exists in at least one of the answers.

- In the "tournament-based approach", GPT-4 is used to select the better reasoning chain as the final solution. Because of the "self-evaluation" ability of LLMs, have you tried to use the majority vote strategy to obtain the final answer as an ablation experiment and compute the accuracy? In GPT-3.5 based approaches, is the "tournament" based on GPT-4 or GPT-3.5 (you mentioned that the GPT-4 is prompted to compare the current chains with (i+1)-th chain (Section 3) )?

---

> ### Author Response · Authors · 2023-11-15
> **Rebuttal Response [1/2]**
>
> We sincerely thank you for your constructive feedback! We address the comments and questions below.
>
> > Evaluation over more datasets, like GSM8K and PRM800K.
>
> Thanks for your suggestion! We have added additional experiments on GSM8K, and our approach continues to achieve better final-answer accuracy. Please refer to the **Common Response** for detailed results.
>
> (Note: The PRM800K also employs the MATH dataset, which is the same dataset our paper's experiments are based on.)
>
> > Is our improvement coming from the hierarchical policy framework or the "self-evaluation" process when choosing the better reasoning chains?
>
> Thanks a lot for your suggestion. We have added two ablation experiments in our revised paper. In the first experiment, we investigate whether the baseline CoT + Sampling performance can be improved through our tournament-based evaluation process. Results are shown in Table A below. We find that adding our tournament selection process to the baseline does not ourperform our approaches that adopt the hierarchical-policy framework, demonstrating that our hierarchical policy plays a significant role in enhancing LLM's ability to solve challenging problems.
>
> In the second experiment, we investigate the effect of majority voting vs. our tournament selection on the final answer accuracy of our hierarchical policy approaches. Results are shown in Table B below. We find that for our hierarchical policy approaches, adopting our tournament selection process yields better final answer accuracy than using majority voting, so our tournament selection process is also helpful. Intuitively, this is because not all of the tactics and hints produced by our high-level leader are helpful, and some of them might mislead the low-level follower, potentially causing it to generate a consistent but wrong answer under a misleading high-level guidance. By evaluating reasoning chains using our tournament-based selection approach, we effectively remove those that exhibit reasoning mistakes and keep those that are more trustful.
>
> **Table A: Effect of our tourament-based reasoning chain selection approach on the CoT + Sampling baseline. For the baseline, we randomly partition the 64 sampled reasoning chains per problem into $n=4$ groups, each having $m=16$ reasoning chains. We use GPT-3.5 as the language model for our low-level follower and the CoT Sampling Baseline.**
> | Method      | Baseline CoT + Majority Voting | Baseline CoT + Tournament  | Ours - Tactic (w/ Tournament)  | Ours - Retrieval (w/ Tournament) |
> |------------------|------|-------------|-------------|-------------|
> | Answer Accuracy  | 22.14 | 22.86   |**25.00**  | **27.85**   |
>
>
> **Table B: Effect of majority voting and our tournament-based reasoning chain selection on the final-answer accuracy of CoT + Sampling baseline and our hierarchical policy approaches. For "Majority Voting", we directly perform majority voting over the 64 sampled reasoning chains per problem. For "Majority Voting over Groups" and "Tournament", we adopt $n=4$ groups, each having $m=16$ reasoning chains. We use GPT-3.5 as the language model for our low-level follower and the CoT Sampling Baseline.**
> | Method |  Baseline CoT | Ours - Tactic | Ours - Retrieval |
> |------------------|------|-------------|-------------|
> | Majority Voting | 22.14 | 21.79 | 23.57 |
> | Majority Voting over Groups | 19.70 | 20.24 | 23.39 |
> | Tournament | 22.86   |**25.00**  | **27.85** |
>
>
> > What's the motivation of the "Grouped-Majority Recall" metric?
>
> The motivation of our Grouped-Majority Recall metric is presented in the second and the fourth paragraph of Sec. 4.1, along with Figure 4. We have also made our motivation more clear in our revised paper.
>
> In detail, the purpose of our metric is to assess the **“visibility”** of the correct answers among solutions generated along the exploration process. A correct answer is “visible” if it not only exists in at least one of reasoning chains, but also occupies a substantial proportion of them, even though it might not be the majority answer. *We find that standard metrics like accuracy and recall do not perfectly align with our goal.* This is because standard recall poorly correlates the prominence of correct answers, and standard accuracy does not reflect the scenario where LLM identifies correct solutions by a reasonable chance (and more than once), but the correct solutions are submerged during majority voting. On the other hand, our Grouped-Majority Recall metric allows correct answers that are identified by a reasonable chance but become obscured in majority voting to emerge, while ensuring that all correct answers recognized by our metric take up the majority in at least one reasoning group and are represented in more than one reasoning chains.

---

> ### Author Response · Authors · 2023-11-15
> **Rebuttal Response [2/2]**
>
> > A more intuitive idea may be the percentage of questions whose ground truth answer exists in at least one of the answers.
>
> The percentage of questions whose ground truth answer exists in at least one of the answers is the "standard recall" metric referred in the last comment. From our analysis, we find that the standard recall metric is not an ideal metric to assess the visibility of correct answers, because as the number of reasoning chain samples increases, the standard recall steadily improves, but the standard accuracy quickly plateaus after a few reasoning chain samples (as shown in Fig. 4a of the paper). Thus, the standard recall metric does not correlate well with the prominence of correct answers.
>
> > In GPT-3.5 based approaches, is the "tournament" based on GPT-4 or GPT-3.5 (you mentioned that the GPT-4 is prompted to compare the current chains with (i+1)-th chain (Section 3) )?
>
> In our tournament-based reasoning chain selection process, we use GPT-4 to compare the current reasoning chain with the $(i+1)$-th chain, where the reasoning chains can be generated by either GPT-3.5 or GPT-4.
>
> We further add an ablation study in Table 5 of the revised paper to compare using GPT-3.5 or GPT-4 to conduct our tournament-based reasoning chain selection process. We observe that GPT-4 demonstrates stronger performance as a reasoning chain selector than GPT-3.5 with a limited number of comparison repetitions (e.g., $k = 1$). Additionally, with only $k=1$ comparison repetition, GPT-4 already leads to a noteworthy enhancement of final answer accuracy over the CoT Sampling baseline.

---

> > ### Author Response · Authors · 2023-11-22
> >
> > Dear Reviewer Yers,
> >
> > Hope this message finds you well. As we are approaching the end of the author-reviewer discussion period, we would like to kindly request your feedback on our rebuttal. Again, we truly appreciate your time and effort reviewing our paper!
> >
> > Best,
> >
> > Authors

---

### Official Review · Reviewer_qX4Z · 2023-11-01

**Soundness:** 3 good
**Presentation:** 2 fair
**Contribution:** 3 good
**Rating:** 5
**Confidence:** 3

**Summary:**

The authors propose a new approach to improve the math reasoning capabilities of Large Language Models (LLMs) by framing them as hierarchical policies. The hierarchical policy consists of two parts: a "visionary leader" and a "follower". The leader suggests multiple high-level problem-solving tactics or hints, while the follower carries out detailed reasoning based on each of these high-level instructions. For each leader's directive, the follower samples multiple reasoning chains to create a group of potential solutions. To select the best solution from these groups, the authors introduce a tournament-based selection method. Experimental results on the MATH dataset show that this approach generates meaningful hints, and improves the accuracy of the final answer on challenging problems.

**Strengths:**

1. I like the idea of generating hints first and then apply low level detailed reasoning. This strategy is intuitive and more like what we humans do in the real life. The authors also provide an effective way of sampling answers based on the hints and new strategy to select the best-of-n.
2. The experiment results show that the proposed method are better than both CoT + self-consistency and ToT + self-consistency baselines.

**Weaknesses:**

1. Generally I am not very sure about the novelty of the proposed method. I am mostly familiar w/ the math reasoning works but not familiar w/ the topic of LLM planning. The novelty may be a weakness; or may not --- I would like to refer to the opinions from other reviewers.
2. Some descriptions of the method/experiment are confusing. Equation (1) and the relevant text is an example. The authors integrate w.r.t. $h$, so they treat $Pr(A|h,Q)$ as a probability density function so the integral $Pr(A|Q)$ should be a probability mass function. Yet the authors use the same notation $Pr$ which is quite confusing. More importantly, although we can understand what the authors would like to express after reading the whole section, the equation itself is invalid as $h$ is a discrete random variable rather than a continuous random variable so you cannot integrate w.r.t. $h$. Please note that mathematical notations in a paper is for helping readers to understand your idea more easily; but Equation (1) is not helping but instead making it even harder to understand. Actually, the key point of the section is just one sentence: "More generally, our strategy samples all the different hints returned by $\pi_{high}$ with equal probabilities.". And this is already clear enough. Similar feelings also appear in the section introducing the "Grouped-Majority Recall" metric, where the authors use quite long paragraphs to explain the details of it but the organization is not quite good and thus make it not easy to get the motivation of proposing this new metric. We generally suggest the authors to improve the expression and organization in these sections for better readability.
3. I may miss it but it seems there is no discussion about the accuracy/effectiveness of the tournament-based selection. IMO, it's non-trivial for LLM to accurately select the best answer among $n$ by iteratively comparing pair by pair. The authors do not provide the reason of introducing this new approach and why not use alternatives like another majority voting over the $n$ candidates. Also, it is possible to make a majority voting over the $n \times m$ candidates directly; the authors do not demonstrate the necessity of the hierarchical selection strategy that majority votes within each group to get $n$ candidates first and select by tournament.

**Questions:**

1. For retrieval-based hints generation, after you find similar examples in the training data, how do you identity the hints for these examples? The original MATH dataset doesn't contain such annotations. Do you generate the hints by LLM? If so, what's the accuracy of these hints?

---

> ### Author Response · Authors · 2023-11-15
> **Rebuttal Response**
>
> We sincerely thank you for your constructive feedback! We address the comments and questions below.
>
> > Is our approach novel? I would like to refer to the opinions from other reviewers.
>
> We'd like to kindly highlight the comments from two reviewers, `vZb3` and `ufPr`, both of whom acknowledge the novelty of our framework and approach.
>
> We’d also like to clarify our paper's contribution and novelty below:
> - Framing large language models (LLMs) as a hierarchical policy that encompass both “high-level” and “low-level” cognitive processes, thereby allowing LLMs to effectively explore expansive solution spaces in complex problems. To our best knowledge, this line of work is underexplored in the field of LLM problem solving.
> - Proposing two effective approaches for the high-level leader policy to generate tactics and hints that guide the low-level follower policy. These approaches include generating concise and relevant techniques and concepts along with retrieving relevant problems.
> - Proposing a novel tournament-based approach to efficiently and effectively select the final reasoning chain.
>
> > Clarification of Equation 1
>
> Thanks for your suggestion. The answer, hint, and question spaces are all discrete, so we have replaced the integral sign with the sum sign and changed our equation (1) to
>
> $\mathrm{Pr}(A|Q) = \sum_{h}\mathrm{Pr}(A|h, Q) \cdot \text{Unif}(h\in H)\cdot \mathrm{Pr}(H|Q)$
>
> in our updated paper.
>
> (As a side note, we originally used the integral sign as we were using a more general measure theory-style notation, under which one could use the integral sign to integrate over any measurable space, such as discrete spaces with counting measure and $\mathbb{R}^n$ with Lebesgue measure. In our original Equation 1, the integral was with respect to the counting measure over our discrete hint space, i.e., $\mathrm{Pr}(A|Q) = \int_{h}\mathrm{Pr}(A|h, Q) \cdot \text{Unif}(h\in H)\cdot \mathrm{Pr}(H|Q) d\mu(h)$ and $\mu$ is the counting measure. In our original paper, we omitted $d\mu(h)$ for brevity, which possibly caused the confusion.)
>
> > Core message of our "Probabilistic Interpretation" Section
>
> Thanks for your suggestion. As suggested, we have highlighted the sentence `our strategy samples all the different hints returned by $\pi_{high}$ with equal probabilities` in our updated paper.
>
> We'd like to also clarify that the purpose of our "Probabilistic Interpretation" section is to provide an intuition of why our proposed hierarchical policy approach endows LLMs with better exploration ability in solving challenging reasoning problems, *compared to previous approaches that directly sampling multiple reasoning chains to explore the solution space*. Our analysis reveals that directly sampling reasoning chains is prone to stucking in a single or a few high-probability tactics, while our hierarchical approach allows tactics to be more uniformly sampled, which is beneficial for solving challenging math problems that require out-of-the-box thinking.
>
>
> > Improve the writing organization for our motivation of "Grouped-Majority Recall"
>
> Thanks for your suggestion. We have revised our paper and added multiple boldface highlights to make the core messages clear when we describe the motivation of our Grouped-Majority Recall metric.
>
> In detail, the goal of our metric is to assess the **“visiblity”** of the correct answers among solutions generated along the exploration process. *A correct answer is “visible” if it not only exists in at least one of reasoning chains, but also occupies a substantial proportion of them, even though it might not be the majority answer.* The second paragraph of Sec. 4.1, along with Figure 3, demonstrates why standard metrics like accuracy and recall are not suitable for our goal. The third paragraph of Sec. 4.1 explains the definition of our Grouped-Majority Recall metric. The fourth paragraph of Sec. 4.1 demonstrates why our Grouped-Majority Recall metric better measures the visibility of correct answers compared to the standard accuracy and recall metrics.
>
>
> > When we select the final answer among the groups of reasoning chains explored by our hierarchical policy, is our tournament-based selection approach better than direct majority voting?
>
> We have added an experiment in our revised paper. We demonstrate that our tournament-based reasoning chain selection approach yields better final-answer accuracy over majority voting. Please see **Common Response** for more details.
>
> > Clarification on retrieval-based hint generation
>
> After the high-level leader retrieves a relevant problem along with its step-by-step solution, the problem and the solution directly serves as the hint for the low-level follower to carry out the problem-solving process (see Table 10 in Appendix for an illustration). There are no further prompting to generate (more concise and high-level) hints using the retrieved problem and solution.

---

> > ### Comment · Reviewer_qX4Z · 2023-11-15
> >
> > Thanks for your prompt response! The comment address most of my concerns in the initial review.
> >
> > To authors:
> > Feel free to comment again if you make further revision to the paper before rebuttal period ends and I would take a look.
> >
> > To AC:
> > My impression of the paper is "quite near the threshold yet towards rejection" when I initially reviewed. After this round of clarification I feel like the paper gets better, but in general still quite near the threshold.

---

> ### Author Response · Authors · 2023-11-15
>
> Dear Reviewer qX4Z,
>
> Thanks for your feedback! Could you please suggest further revisions (if any) that would make our work more satisfactory to you and potentially improve our rating in your evaluation? Your guidance would be greatly appreciated.

---

> > ### Comment · Reviewer_qX4Z · 2023-11-15
> >
> > Hi authors,
> >
> > For me, what makes me feel like this is a threshold work is mainly due to the innovation of the method/approach itself. The method is novel and shows improvement on certain tasks, but from my personal opinion is not enough to like "wow this is really nice and decent". For the revision period I encourage to better optimize the expression and try to better highlight the most important motivations and key contributions in your approach; currently it is not very clear when taking a first glance of paper and people have to carefully read line by line to fully understand your idea. A good presentation would catch readers' eyes at the first glance and lead the readers thorough the paper :) In general I think this is a not-so-bad paper and I may raise up to 6 --- but tbh I don't think 5/6 makes too much difference in the decision though. I would definitely try to do my best to provide a responsible opinion through the process; please just believe your effort would pay back as long as you work hard :)

---

> ### Author Response · Authors · 2023-11-16
>
> Thanks for your feedback! We've revised the introduction and methodology sections, using bold and italic formatting to emphasize our core motivations and key contributions more clearly and directly. We hope that combined with our revisions in the later experiment and analysis sections, we have enhanced the overall clarity and understanding for our readers. Should you have any further suggestions, please let us know.

---

> > ### Author Response · Authors · 2023-11-22
> >
> > Dear Reviewer qX4Z,
> >
> > Hope this message finds you well. As we are approaching the end of the author-reviewer discussion period, we would like to kindly request your feedback on our latest rebuttal updates. Again, we truly appreciate your time and effort reviewing our paper!
> >
> > Best,
> >
> > Authors

---

> > > ### Comment · Reviewer_qX4Z · 2023-12-04
> > >
> > > Thanks for all the effort during the rebuttal period. After reviewing all the changes and the other feedback and discussions from other reviewers and AC, I decide to keep my current score.

---

### Official Review · Reviewer_vZb3 · 2023-11-01

**Soundness:** 2 fair
**Presentation:** 2 fair
**Contribution:** 2 fair
**Rating:** 3
**Confidence:** 4

**Summary:**

The study presents an innovative framework that uses a hierarchical policy structure to improve problem-solving in Large Language Models (LLMs). A "low-level follower" executes details while a high-level "visionary leader" generates strategies. Using a tournament-based solution selection process, the authors' approach, which was evaluated on the MATH dataset, demonstrates improved strategy exploration and answer accuracy.

**Strengths:**

1. Innovative Framework: The hierarchical policy framework for problem-solving is a novel approach that capitalizes on the creative potential of LLMs.

2. Sophisticated Selection Mechanism: The tournament-based selection process is a unique and strategic method to pinpoint the most effective solution, which mimics evolutionary selection processes to optimize problem-solving outcomes.

**Weaknesses:**

1. Limited Dataset Representation: The study's findings are primarily based on the MATH dataset, which raises concerns about the model's performance on other datasets that present different challenges, such as the GSM8k. This limits the understanding of the model's adaptability and effectiveness across various types of reasoning tasks.

2. Ambiguity in Model Details: The paper does not specify which versions of GPT-3.5 and GPT-4 were used, nor does it detail the hyper-parameters involved. Such information is critical for replicating the study.

3. Cost Analysis Omission: There is no comprehensive analysis of the computational costs associated with different methods, including the number of tokens generated and encoded. Such an analysis is essential to evaluate the model's efficiency and practicality.

**Questions:**

How does the model perform on other representative datasets like GSM8k, and can you provide comparative analysis to demonstrate its versatility across various domains?

Could you specify the versions and hyper-parameters of GPT-3.5 and GPT-4 used in your experiments, and discuss how different configurations might affect the model's problem-solving capabilities?

Can you provide a detailed cost analysis, including the number of generation tokens and encoding tokens required, to better understand the computational efficiency of your proposed method compared to traditional approaches?

---

> ### Author Response · Authors · 2023-11-15
> **Rebuttal Response [1/2]**
>
> We sincerely thank you for your constructive feedback! We address the comments and questions below.
>
> > Evaluation over GSM8k
>
> Thanks for your suggestion! We have added additional experiments on GSM8K, and our approach continues to achieve better final-answer accuracy. Please refer to the **Common Response** for detailed results.
>
> > Version of GPT-3.5/GPT-4 along with their hyperparameters
>
> We use GPT-3.5-turbo-0613 and GPT-4-0613 for all the experiments. For hyperparameter details, we set the decoding temperature to 0.3 for tournament-based reasoning chain selection and 0.7 otherwise (i.e., for hint generation from the high-level leader, reasoning chain generation from the low-level follower, along with the baselines). These hyperparameters were briefly mentioned in the first paragraph of our experiment setup. For clarity purposes, we have added an additional section (Appendix B in the updated PDF) to describe the GPT-3.5/4 settings used in the paper.
>
>
> > Effect of Different Model Configurations
>
> Thanks for your suggestion. We have added an experiment in Table 5 and 6 of the revised paper, where we perform ablations on different models (GPT-3.5 / GPT-4), different comparison repetitions $k$, and different temperatures $T$ used to perform our tournament-based reasoning chain selection process (in our previous paper's experiments, we used GPT-4 model, $k=1$, and $T=0.3$ for tournament selection). We copy the results below for convenience. We find that GPT-4 demonstrates stronger performance as a reasoning chain selector than GPT-3.5 with a limited number of comparison repetitions (e.g., $k = 1$). In particular, with only $k=1$ comparison repetition, GPT-4 already leads to a noteworthy enhancement of final answer accuracy over the CoT Sampling baseline. Additionally, when $0<T<1$, different temperatures during tournament selection have little effect on the final answer accuracy.
>
> **Table 5: Ablation on using different models to conduct our tournament-based reasoning chain selection process, along with using different k, i.e., different numbers of comparison repetitions, during this process. We compare our retrieval-based method with the CoT Sampling baseline. We use GPT-3.5 as the language model for our low-level follower and the CoT Sampling Baseline.**
> | Method                | Ours - Retrieval | Ours - Retrieval | Ours - Retrieval | Ours - Retrieval | Ours - Retrieval | Ours - Retrieval | CoT Sampling + Voting |
> |-----------------------|------------------------|------------------------|------------------------|------------------------|------------------------|------------------------|-----------------------|
> | **Tournament Model**      | **GPT-3.5**                | **GPT-3.5**                | **GPT-3.5**                | **GPT-4**                  | **GPT-4**                  | **GPT-4**                  | **N/A**                   |
> | **#Comparisons**         | $k=1$                  | $k=3$                  | $k=5$                  | $k=1$                  | $k=3$                  | $k=5$                  | N/A                   |
> | Answer Accuracy       | 21.43                  | 22.86                  | **27.14**              | **27.85**              | **27.85**              | **27.85**              | 22.14                 |
>
> **Table 6: Ablation on using different temperature (T) in our tournament-based reasoning chain selection process. Results are obtained using GPT-4 as our tournament selection model with $k=1$ comparison repetition. We use GPT-3.5 as the language model for our low-level follower and the CoT Sampling Baseline.**
> | Temperature      | $T=0$  | $T=0.3$       | $T=0.7$       | $T=1.0$       |
> |------------------|------|-------------|-------------|-------------|
> | Answer Accuracy  | 26.43 | **27.85**   | 27.14   | 27.14   |

---

> ### Author Response · Authors · 2023-11-15
> **Rebuttal Response [2/2]**
>
> > Cost analysis comparison.
>
> Thanks for your suggestion. We compare the cost between our approach and the CoT + Sampling Baseline below (unit in dollars; tournament performed with GPT-4; a total of 64 reasoning chains are generated per question). We also add this table in Appendix Section C. While our approach is slightly more expensive than the CoT baseline when GPT-3.5 serves as the low-level follower, it is more cost efficient than the baseline when GPT-4 serves as the follower, and moreover yields better final answer accuracy.
>
>
> | **Method** | CoT + Sampling Baseline | Ours - Tactics | Ours - Retrieval | CoT + Sampling Baseline | Ours - Tactics | Ours - Retrieval |
> |-------------------|---------------|-------------------------------|---------------------|-------------|----------------------------|-------------------|
> | **Low-Level Follower** | GPT-3.5 | GPT-3.5 | GPT-3.5 | GPT-4 | GPT-4 | GPT-4|
> | **Sampling Cost**      | 10.72           | 11.71            | 9.83                | 204.16        | 191.11         | 188.47           |
> | **Tournament Cost**    | None           | 2.42                          | 4.45                | None         | 4.52                       | 5.92             |
> | **Total Cost**         | 10.72           | 14.12            | 14.28               | 204.16        | 195.63         | 194.40           |
>
> Additionally, we also compare the average number of input and output tokens between the baseline and our approach. The number of output tokens is calculated over the sum of 64 reasoning chains sampled per question. (For both GPT-3.5 and GPT-4, input tokens cost half as much as output tokens.)
>
> | **Method** | CoT + Sampling Baseline | Ours - Tactics | Ours - Retrieval | CoT + Sampling Baseline | Ours - Tactics | Ours - Retrieval |
> |--------------------|-----------------|------------------|---------------------|---------------|----------------|-------------------|
> | **Low-Level Follower** | GPT-3.5 | GPT-3.5 | GPT-3.5 | GPT-4 | GPT-4 | GPT-4|
> | Avg. # Input Tokens Per Question (including tournament in our approach) | 0.11K           | 0.73K            | 1.88K               | 0.11K         | 0.88K          | 2.00K      |
> | Avg. # Output Tokens Per Question (including tournament in our approach)| 38.20K          | 41.61K           | 34.32K              | 24.25K        | 22.85K         | 22.14K            |

---

> > ### Author Response · Authors · 2023-11-22
> >
> > Dear Reviewer vZb3,
> >
> > Hope this message finds you well. As we are approaching the end of the author-reviewer discussion period, we would like to kindly request your feedback on our rebuttal. Again, we truly appreciate your time and effort reviewing our paper!
> >
> > Best,
> >
> > Authors

---

### Author Response · Authors · 2023-11-15
**Common Response**

We sincerely thank all reviewers and ACs for their constructive feedback! We're glad that reviewers found our approach novel (`vZb3, ufPr`), intuitive (`qX4Z, ufPr`), strategic (`vZb3`), and effective (`qX4Z, Yers, ufPr`); our experiment results interesting (`Yers`) and extensive (`ufPr`); and our paper well-written (`Yers, ufPr`). We address common reviewer questions below, and we address individual questions under reviewer comments. We have also updated our paper with revisions in red.

> Evaluation over other datasets like GSM8K

We have added results on GSM8K in the table below as well as our updated paper. We adopt GPT-3.5 as the low-level follower, and we sample 32 reasoning chains per problem. For "Majority Voting", we directly perform majority final-answer voting over the 32 reasoning chains. For tournament, we adopt $n=4$ groups, each having $m=8$ reasoning chains.

We find that while GSM8k features easier mathematics problems than the MATH Level-5 questions used in our paper's experiments, our approach continues to achieve better final-answer accuracy.

| Method | CoT + Sampling Baseline, Majority Voting| CoT + Sampling Baseline, Tournament | Ours - Retrieval (w/ Tournament) |
|---|---|---|---|
| Accuracy | 88.45   | 88.85 | **89.91** |

> When we select the final answer among the groups of reasoning chains explored by our hierarchical policy, is our tournament-based selection approach better than direct majority voting?

We have added an ablation experiment in our revised paper. In this experiment, we investigate the effect of majority voting vs. our tournament selection process on the final answer accuracy of our hierarchical policy approaches along with the baseline. Results are shown in the table below. We find that
- For our hierarchical policy approaches, adopting our tournament-based reasoning chain selection process yields better final answer accuracy than using majority voting, demonstrating the effectiveness of our tournament selection process. Intuitively, this is because not all of the tactics and hints produced by our high-level leader are helpful, and some of them might mislead the low-level follower, potentially causing it to generate a consistent but wrong answer under a misleading high-level guidance. By evaluating reasoning chains using our tournament-based selection approach, we effectively remove those that exhibit reasoning mistakes and keep those that are more trustful.
- Additionally, for the CoT + Sampling baseline, adopting our tournament-based reasoning chain selection process does not ourperform our approaches that employ the hierarchical-policy framework. This demonstrates that our hierarchical policy plays a significant role in enhancing LLM's ability to solve challenging problems.

**Table: Effect of majority voting and our tournament-based reasoning chain selection on the final-answer accuracy of CoT + Sampling baseline and our hierarchical policy approaches. For "Majority Voting", we directly perform majority voting over the 64 sampled reasoning chains per problem. For "Majority Voting over Groups" and "Tournament", we adopt $n=4$ groups, each having $m=16$ reasoning chains. We use GPT-3.5 as the language model for our low-level follower and the CoT Sampling Baseline.**
| Method |  Baseline CoT + Sampling | Ours - Tactic | Ours - Retrieval |
|------------------|------|-------------|-------------|
| Majority Voting | 22.14 | 21.79 | 23.57 |
| Majority Voting over Groups | 19.70 | 20.24 | 23.39 |
| Tournament | 22.86   |**25.00**  | **27.85** |

---

### Meta-Review · Area_Chair_jnYL · 2023-12-06

**Metareview:**

This paper introduces a hierarchical approach for sampling from LLMs to solve reasoning tasks. The approach prompts the LLM as a top-level expert to provide $n$ domain-specific strategies, each of which are used to generate $m$ outputs. A tournament approach is then used to select the best output, by first taking a majority vote within each of the $n$ top-level groups, randomly selecting an output in each group matching the majority vote response for that group, and then having the LLM evaluate which of the selected outputs is better via pairwise comparison for adjacent outputs, when the outputs are indexed by their top-level group index. The authors also propose a separate, retrieval-based approach that first retrieves related examples from a database of example embeddings.

The hierarchical approach taken in this paper is interesting, but as the reviewers pointed out, many of the design decisions are not properly motivated or ablated. For example, the tournament selection is not properly compared to simple majority voting or CoT with a similar tournament selection approach (which importantly benefits from a pairwise evaluation conducted by the LLM). Many reviewers also pointed out that the evaluation datasets are limited, and this paper will benefit from evaluating the proposed method on additional datasets.

Similarly, the retrieval-based method seems quite unrelated to the main approach described in the work, and the design space of the retrieval-based method is not sufficiently explored.

Additionally, the results are based on GPT models from OpenAI, which are blackboxed, making the results not reproducible.

Lastly, the results on the datasets used for evaluation seem mixed. The benefits are not obvious, especially given the additional complexity introduced by the method.

**Justification For Why Not Higher Score:**

This paper is missing proper motivations and ablations for the design choices in the work. The results are not compelling, given the additional complexity introduced by the method. The experiments are also based on GPT models, making the results not guaranteed to be reproducible in the future. It is also not clear what system the experiments are even testing, given the transient nature of the underlying GPT models, which can be updated anytime, and whose implementation details are unknown.

**Justification For Why Not Lower Score:**

N/A

---

### Decision · Program_Chairs · 2024-01-16

Reject